# The mechanism of a high-affinity allosteric inhibitor of the serotonin transporter

Per Plenge[1,7], Ara M. Abramyan[2,7], Gunnar Sørensen [3], Arne Mørk[3], Pia Weikop [4], Ulrik Gether [5], Benny Bang-Andersen [3,6✉], Lei Shi [2✉] & Claus J. Loland [1✉]

The serotonin transporter (SERT) terminates serotonin signaling by rapid presynaptic reuptake. SERT activity is modulated by antidepressants, e.g., S-citalopram and imipramine, to alleviate symptoms of depression and anxiety. SERT crystal structures reveal two S-citalopram binding pockets in the central binding (S1) site and the extracellular vestibule (S2 site). In this study, our combined in vitro and in silico analysis indicates that the bound S-citalopram or imipramine in S1 is allosterically coupled to the ligand binding to S2 through altering protein conformations. Remarkably, SERT inhibitor Lu AF60097, the first high-affinity S2-ligand reported and characterized here, allosterically couples the ligand binding to S1 through a similar mechanism. The SERT inhibition by Lu AF60097 is demonstrated by the potentiated imipramine binding and increased hippocampal serotonin level in rats. Together, we reveal a S1-S2 coupling mechanism that will facilitate rational design of high-affinity SERT allosteric inhibitors.

[1] Laboratory for Membrane Protein Dynamics. Department of Neuroscience, Faculty of Health and Medical Sciences, University of Copenhagen, Copenhagen, Denmark. [2] Computational Chemistry and Molecular Biophysics Unit, Molecular Targets and Medications Discovery Branch, National Institute on Drug Abuse—Intramural Research Program, National Institutes of Health, Baltimore, MD, USA. [3] Lundbeck Research, H. Lundbeck A/S, Copenhagen, Denmark. [4] Laboratory of Neuropsychiatry, Psychiatric Centre Copenhagen, University of Copenhagen, Copenhagen, Denmark. [5] Department of Neuroscience, Faculty of Health and Medical Sciences, University of Copenhagen, Copenhagen, Denmark. [6] Department of Drug Design and Pharmacology, University of Copenhagen, Copenhagen, Denmark. [7] These authors contributed equally: Per Plenge, Ara M. Abramyan. ✉email: BAN@Lundbeck.com; lei.shi2@nih.gov; cllo@sund.ku.dk

Serotonin (5-HT) transmission is involved in many basic brain functions, such as regulation of mood, sleep, appetite, and sexual drive[1]. The serotonin transporter (SERT) is embedded in the presynaptic membrane and terminates 5-HT transmission by rapid reuptake of released 5-HT. SERT belongs, together with the transporters for the neurotransmitters dopamine, norepinephrine, γ-aminobutyric acid, and glycine, to the family of neurotransmitter:sodium symporters (NSSs) that all exploit the electrochemical potential stored in the $Na^+$ gradient to translocate substrates against their concentration gradients[2–4]. Pharmacological inhibition of SERT is known to alleviate symptoms of depression and anxiety, the two most prevalent psychiatric disorders ranking among the top five leading causes of disability worldwide[5,6]. Specifically, the selective serotonin reuptake inhibitors (SSRIs), such as S-citalopram (S-CIT) (Fig. 1a), sertraline and paroxetine, are currently used to treat depression, anxiety, obsessive-compulsive disorder (OCD), and post-traumatic stress disorder (PTSD) among others[7]. The tricyclic antidepressants, such as imipramine (IMI) (Fig. 1a) and amitriptyline, and the multimodal antidepressants vilazodone and vortioxetine[8], also target SERT. In addition, SERT is in part targeted by (illicit) psychostimulants such as MDMA (ecstasy), ibogaine, cocaine, and amphetamine[9,10].

The structures of human SERT[11], drosophila dopamine transporter[12] and two bacterial NSSs, LeuT[13] and MhsT[14], have revealed a conserved structural fold for NSSs with a primary ligand binding (S1) site located in the center of the transmembrane domain. Interestingly, the existence of an allosteric binding site in SERT was reported more than three decades ago[15]. The key observation was that certain SERT inhibitors could impede the dissociation of a pre-bound radiolabeled ligand[16–19]. The most potent impedance being by S-CIT on [³H]S-CIT dissociation, however, only displays a low potency (IC₅₀ is ~5 μM), and in spite of intensive investigations[20,21] no other compound has shown to possess higher potency[20,22]. Based on computational modeling and experimental binding studies, we previously located the low-affinity second binding site for S-CIT and clomipramine to the extracellular vestibule (EV), the entry pathway toward the S1 site[21]. Interestingly, in one of the recent hSERT crystal structures (PDB 5I73), two S-CIT molecules are bound to the protein: one in the S1 site (denoted as S1:S-CIT) and another bound to a binding site in the EV—the S2 site (denoted as S2:S-CIT) ~13 Å above the S1 site[11,23]. This is consistent with previous findings that the EV of LeuT harbors a S2 site capable of binding ligands[24–26]. The comparison of the 5I73 structure to the structure bound with only one S-CIT in S1 (PDB 5I71) shows that they have identical conformations in the EV, and it is not clear whether ligand bindings in the S1 and S2 sites allosterically interact with each other through modulation of any specific structural motif[27,28], which may result in conformational changes.

Allosteric modulators can potentially possess higher selectivity due to the divergence of the binding sites among homologous proteins[29,30] resulting in fewer side effects. In addition, compared with competitive inhibitors to the endogenous ligand, they may retain some of the functions of the target proteins. Indeed, several well-known allosteric modulators possess novel pharmacologic properties such as use-dependency (e.g., lidocaine), or activity modulation of the endogenous ligand (e.g., benzodiazepines), or to perform asymmetric signaling as in metabotropic glutamate receptor complexes[31], all providing advantageous therapeutic potentials over orthosteric modulators. For S-CIT, it has been proposed that its allosteric binding in SERT contributes to its higher efficacy and faster onset observed in clinical trials as compared with racemic citalopram[32–36]. However, the low affinity of S-CIT to S2 relative to S1 hampers the possibility to reveal the specific therapeutic potentials in targeting this site. Thus, a high affinity and selective S2-bound ligand would facilitate not only a thorough mechanistic understanding of allosteric communications between the S2 and S1 sites, but also a proper evaluation of the therapeutic potential of allosteric modulation in SERT.

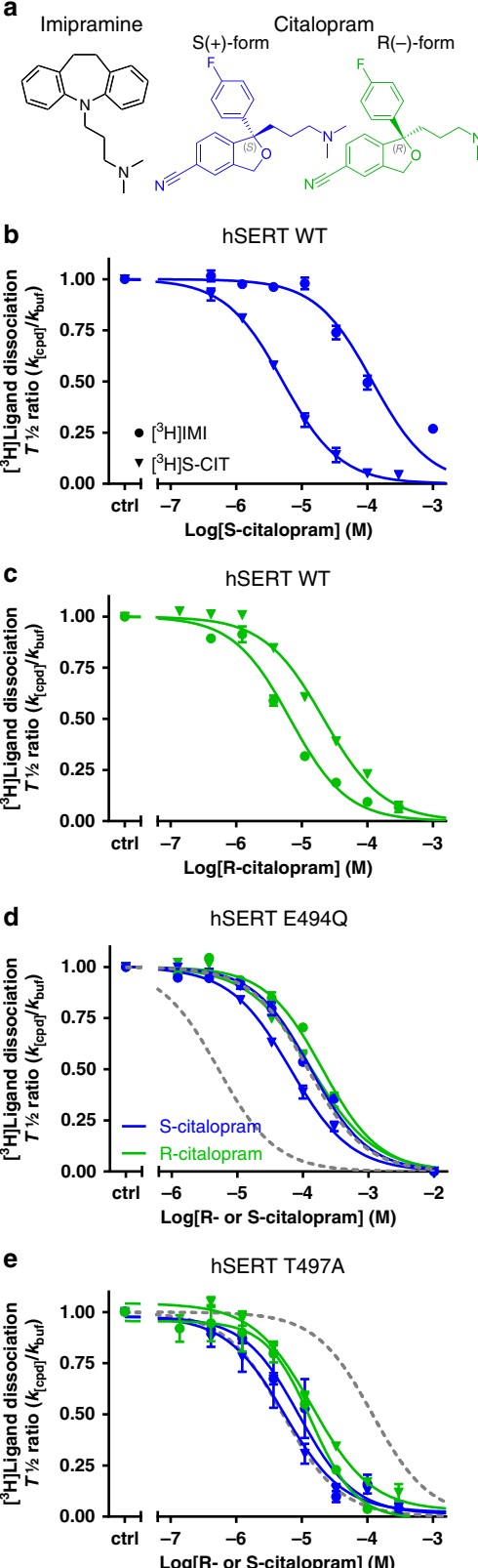

**Fig. 1 Experimental evidence for allosteric binding between the S1 and S2 sites in SERT. a** Chemical structure of the tested drugs. Left: imipramine (IMI). Right: R(−)- and S(+)-citalopram (R-CIT and S-CIT, respectively). **b** Allosteric potency of S-CIT measured as concentration-dependent inhibition of either [3H]IMI or [3H]S-CIT dissociation, prebound to SERT prior to the addition of S-CIT in the indicated concentrations. Data plotted as inhibition of [3H]ligand dissociation rate by S-CIT relative to no S-CIT added (ctrl). Prebound [3H]S-CIT (triangles) results in a 29-fold increase in S-CIT allosteric potency relative to prebound [3H]IMI (circles), with $IC_{50} = 5.1$ [4.6, 5.8] μM and 152 [125, 185] μM, respectively (mean [S.E. interval], $n =$ 3). **c** Allosteric potency of R-CIT measured as in (**b**). Here R-CIT possesses a fourfold higher allosteric potency for prebound [3H]IMI relative to prebound [3H]S-CIT, with $IC_{50} = 5.3$ [4.9; 5.8] μM and 21.5 [20.3; 22.7] μM, respectively (mean [S.E. interval], $n = 3$–6). **d** Allosteric potency of S-CIT (blue) and R-CIT (green) in SERT E494Q predicted to dissipate the allosteric interaction between S1 and S2 sites. The allosteric potency is now collapsed around the observed allosteric potency for S-CIT inhibition of [3H]IMI dissociation (right dotted line) in SERT WT. The allosteric potency of S-CIT (blue) for inhibition of [3H]S-CIT (triangles) and [3H]IMI (circles) dissociation in E494Q is 64.8 [56.6; 74.1] μM and 131 [123; 140] μM, respectively (mean [S.E. interval], $n = 5$–6). The same values for R-CIT are 137 [118; 158] μM and 199 [191; 207] μM, respectively (mean [S.E. interval], $n = 5$). Left dotted line is allosteric potency for S-CIT with prebound [3H]S-CIT from (**b**), shown for comparison. **e** Allosteric potency for S-CIT with prebound [3H]S-CIT (triangles) or [3H]IMI (circles) in SERT T497A. Thr497 is predicted to mediate the S1:S2 allosteric interaction (see Fig. 2). The $IC_{50}$ is collapsed, around the allosteric potency for [3H]S-CIT dissociation from SERT WT (dotted lines); $IC_{50}$ for S-CIT: 5.10 [3.20; 8.00] μM and 8.80 [5.90; 13.4] μM for inhibition of [3H]S-CIT and [3H]IMI dissociation, respectively. The same values for R-CIT are 17.8 [15.7; 20.2] and 11.6 [10.1; 13.3], respectively (mean [S.E. interval], $n = 3$). Source data are provided as a Source Data file.

Here, we provide mechanistic evidence for allosteric modulations between the S1 and S2 sites in SERT by showing that the effects of ligand binding to the S1 site allosterically propagate through altered conformation of a structural motif between the S1 and S2 sites. In the context of these findings, we report the identification and characterizations of a high-affinity S2-bound inhibitor for SERT, Lu AF60097 ((S)-1-(4-fluorophenyl)-1-(3-(4-(2-oxo-1,2-dihydroquinolin-7-yl)piperidin-1-yl)propyl)-1,3-dihydroisobenzofuran-5-carboxamide), in vitro, in silico, and in vivo.

## Results

**S1-binding is allosterically connected to S2-binding.** We have previously shown that the binding of either S-CIT or clomipramine to the S2 site inhibits [3H]S-CIT dissociation from the S1 site[21]. To examine whether ligand binding to S1 would induce conformational changes of SERT that allosterically affect ligand binding to S2[37], we first evaluated whether the inhibitory potency of a S2-bound ligand can be differentially affected by the identity of the S1-bound ligand. For S-CIT and IMI, we specifically assumed that they would bind exclusively to S1 at nanomolar concentrations (denoted as S1:S-CIT and S1:IMI). Accordingly, we added 25 nM of either [3H]S-CIT or [3H]IMI to membranes of COS-7 cells transiently expressing hSERT wild type (WT), and then measured the dissociation rates of the two ligands in the presence of increasing concentrations of S- or R-CIT (0.4 μM–1 mM) that should occupy the S2 site at high concentrations (S2:S-CIT and S2:R-CIT). The increase in S2-occupancy by either S- or R-CIT results in a dose-dependent inhibition of dissociation by the S1-bound radioligand. The concentration of a S2-bound ligand causing 50% decrease in the dissociation of a S1-bound radioligand ($IC_{50}$) is used as a measure to reflect the inhibitory (allosteric) potency for S2 binding (see "Methods"). We found

that the allosteric potency of S2:S-CIT was 29-fold higher in the presence of S1:[3H]S-CIT than in the presence of S1:[3H]IMI (Fig. 1b, Table 1). In contrast, the allosteric potency of S2:R-CIT was reversed, i.e., lower in the presence of S1:[3H]S-CIT relative to S1:[3H]IMI (Fig. 1c, Table 1). The results indicate that the allosteric potency of a S2-bound ligand is sensitive to the identity of the S1-bound ligand. Thus, assuming no direct interaction between the S1- and S2-bound ligands (see below), these results support the idea that SERT conformational changes induced by ligand binding to S1 modulate ligand binding to S2.

To probe the mechanistic details of a possible allosteric interaction between ligands bound to S1 and S2, we performed extensive molecular dynamics (MD) simulations at microsecond scale using the ts3 and WT SERT models (see "Methods") in complexes with different combinations of S1- and S2-bound ligands (Fig. 2, Supplementary Table 1). We first compared the resulting conformations of the simulations in the presence of S1: S-CIT without any ligand bound in S2 (denoted as S1:S-CIT/S2: apo) to that of the 5I71 structure. When comparing the two S-CIT bound SERT crystal structures, 5I71 and 5I73, we found that the EV space occupied by the S2:S-CIT in 5I73 is filled by a dodecane in 5I71 (likely part of a lipid molecule used in the crystallization process) (Supplementary Fig. 1). Interestingly, in the absence of any ligand in S2, our simulations in both ts3 and WT constructs showed that the side chains of Phe334 and Phe335 of TM6 move toward the center of the EV and form an aromatic cluster with Phe556 that rotates inward (Supplementary Fig. 1). Consequently, Phe335 and Phe556 occupy the space that overlaps with dodecane and S2:S-CIT in the crystal structures (Supplementary Fig. 2). On the other hand, the resulting EV conformations of the S1:S-CIT/S2:S-CIT simulations were similar to that of the 5I73 structure. Thus, the conformational differences observed between S1:S-CIT/S2:apo and S1:S-CIT/S2:S-CIT conditions suggest that the binding of S-CIT in S2 is associated with robust conformational rearrangements.

We then compared the SERT conformations in the S1:S-CIT/S2:apo and S1:IMI/S2:apo conditions (Fig. 2b, d). We found that different moieties of these two S1 ligands that face TM10, i.e., the cyano group of S-CIT and the aromatic ring of IMI, have significantly different impacts on the conformation of the bulge helical turn in TM10 (Leu492 to Thr497). In particular, we found a remarkable difference in the $\chi_1$ dihedral angle of Thr497 depending on the docked compound. The cyano group of S1:S-CIT favors the $\chi_1$ rotamer of Thr497 to be in gauche−, while this rotamer is more likely in gauche+ in the presence of S1:IMI (Fig. 2g).

Next, we built and equilibrated SERT WT models in the S1:S-CIT/S2:S-CIT and S1:IMI/S2:S-CIT conditions (see "Methods", Fig. 2c, e). The analysis of the MD simulations of these conditions showed that the $\chi_1$ rotamer of Thr497 in the presence of S1:S-CIT is further stabilized in gauche− by the addition of S2:S-CIT, whereas the S2:S-CIT in the same pose is not stable in the presence of S1: IMI, forcing Thr497 in the latter condition to rotate from gauche+ to the gauche− rotamer (Fig. 2f). When Thr497 is in gauche−, we found that the S1-gating residue, Phe335, cannot form a stable interaction with the benzofuran moiety of S2:S-CIT and transitions between gauche− and trans rotamer in the presence of S1:IMI, whereas this interaction is stable in the presence of S1:S-CIT. To quantify this difference, we counted the numbers of transitions in each condition, and found Phe335 transitions between gauche− and trans rotamer at a rate of 145.4/μs in S1:IMI/S2:S-CIT, while only 1.1/μs in S1:S-CIT/S2:S-CIT.

Thr497 and Phe335 are situated in between the S1 and S2 sites. Their varied configurations in the S1:S-CIT/S2:S-CIT and S1:IMI/ S2:S-CIT conditions correlate with the conformation of Glu494, which shows a higher propensity to form a salt bridge with the

**Table 1 S1 affinity and allosteric potency of investigated compounds to SERT WT.**

| Compound | Inhibition of [3H]S-CIT binding (IC50 in nM) | n | Allosteric potency inhibition of [3H]S-CIT dissociation (IC50 in nM) | n | Allosteric potency inhibition of [3H]IMI dissociation (IC50 in nM) | n |
|---|---|---|---|---|---|---|
| S-citalopram | 5.4 [5.1; 5.7] | 6 | 5200 [4600; 5800] | 3 | 152000 [125000; 185000] | 3 |
| R-citalopram | 180 [150; 210] | 7 | 21500 [20300; 22700] | 6 | 5340 [4880; 5830] | 3 |
| Lu AF60097 | 270 [200; 360] | 6 | 6500 [5070; 8310] | 4 | 31 [25; 39] | 6 |
| AE | 1040 [900; 1200] | 3 | 33800 [30700; 37300] | 3 | 120 [100; 140] | 4 |
| AF | 79 [61; 102] | 4 | 10400 [8450; 12900] | 5 | 190 [170; 220] | 3 |

Data for [3H]S-CIT binding inhibition are obtained by non-linear regression analysis of competition binding experiments between [3H]S-CIT and increasing concentrations of the indicated compound (10 concentrations in triplicates). Allosteric potencies are the IC50 from non-linear regression analysis of the change in dissociation rate constant relative to no compound present ($k_{[cmpd]}/k_{buffer}$) as a function of the added compound concentration (Log[cmpd]). Experiments performed on membrane preparations of COS-7 cells expressing SERT WT. Data are shown as mean [SE interval] and are calculated from pIC50 and the SE interval from pIC50 ± SE.

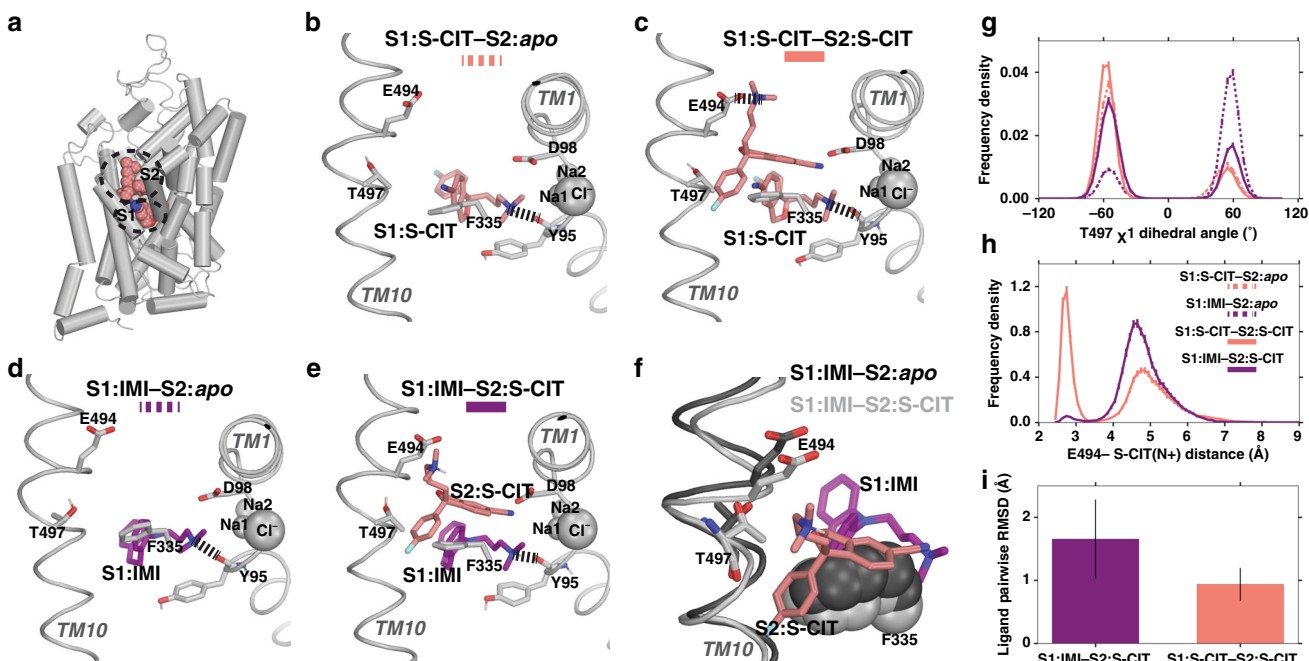

**Fig. 2 MD simulations of the allosteric interaction between SERT S1 and S1 sites.** In the presence of S1:S-CIT the Thr497 χ1 dihedral is mostly shifted towards *gauche−*, whereas in the presence of S1:IMI, it is in gauche+, which in turn affects the salt bridge interaction between S2:S-CIT and Glu494. In all panels, the S1:S-CIT conditions are colored in salmon, whereas the S1:IMI conditions are in purple. **a** A zoomed-out view of the 5I73 structure showing the S1 and S2 sites. **b** A zoomed-in view of the equilibrated model of WT S1:S-CIT/S2:*apo*, **c** S1:S-CIT/S2:S-CIT, **d** S1:IMI/S2:*apo*, and **e** S1:IMI/S2:S-CIT. **g** Distribution of the Thr497 χ1 rotamer for S1:S-CIT/S2:*apo*, S1:IMI/S2:*apo* (dotted lines), and S1:S-CIT/S2:S-CIT and S1:IMI/S2:S-CIT (solid lines) conditions. **h** Distribution of the Glu494/S2:S-CIT distance (minimum distance between the charged N of S2:S-CIT and the two carboxyl oxygens of Glu494) for S1:S-CIT/S2:S-CIT and S1:IMI/S2:S-CIT conditions. **i** S2:S-CIT is more stable in the presence of S1:S-CIT (salmon) than in the presence of S1: IMI (purple) measured by pairwise ligand RMSDs (see "Methods").

charged N of S2:S-CIT in the presence of S1:S-CIT compared with in the presence of S1:IMI (Fig. 2h), resulting in a more stable pose in the former condition (Fig. 2i). Thus, we hypothesized that the observed S2:S-CIT affinity difference in these two conditions (Fig. 1b) is likely resulted from the different impacts of S1-bound ligands on the interaction between S2:S-CIT and Glu494, which are mediated by the Thr497-Phe335 motif.

To experimentally test this hypothesis, we removed the negative charge of Glu494 by the E494Q mutation and measured the allosteric potency of R- and S-CIT in the presence of S1:[3H]S-CIT or [3H]IMI (Fig. 1d). Remarkably, compared with WT, in hSERT E494Q, the allosteric potency of S2:S-CIT was significantly reduced for S1:[3H]S-CIT but did not change for [3H]IMI, and the two potencies became virtually the same. The same was observed for S2:R-CIT (Fig. 1d, Supplementary Table 2). As

our simulation results suggest that Thr497 and Phe335 are sterically crowded in the S1:IMI/S2:S-CIT condition, we further hypothesized that by mutating Thr497 to a residue with a smaller sidechain, the space in between S1 and S2 would be less crowded, which might facilitate the S2:S-CIT binding. Indeed, in the presence of [3H]IMI, the allosteric potency of S2:S-CIT was increased 17-fold in SERT T497A relative to SERT WT. In contrast, the allosteric potency of S2:S-CIT or S2:R-CIT in the presence of [3H]S-CIT was not affected by T497A (Fig. 1e, Supplementary Table 2). Taken together, we propose that Thr497-Phe335 represents a structural motif mediating allosteric communication between the S1 and S2 sites.

**Identification of a high affinity S2 inhibitor.** Based on these findings, we hypothesized that the binding of S2-ligands having

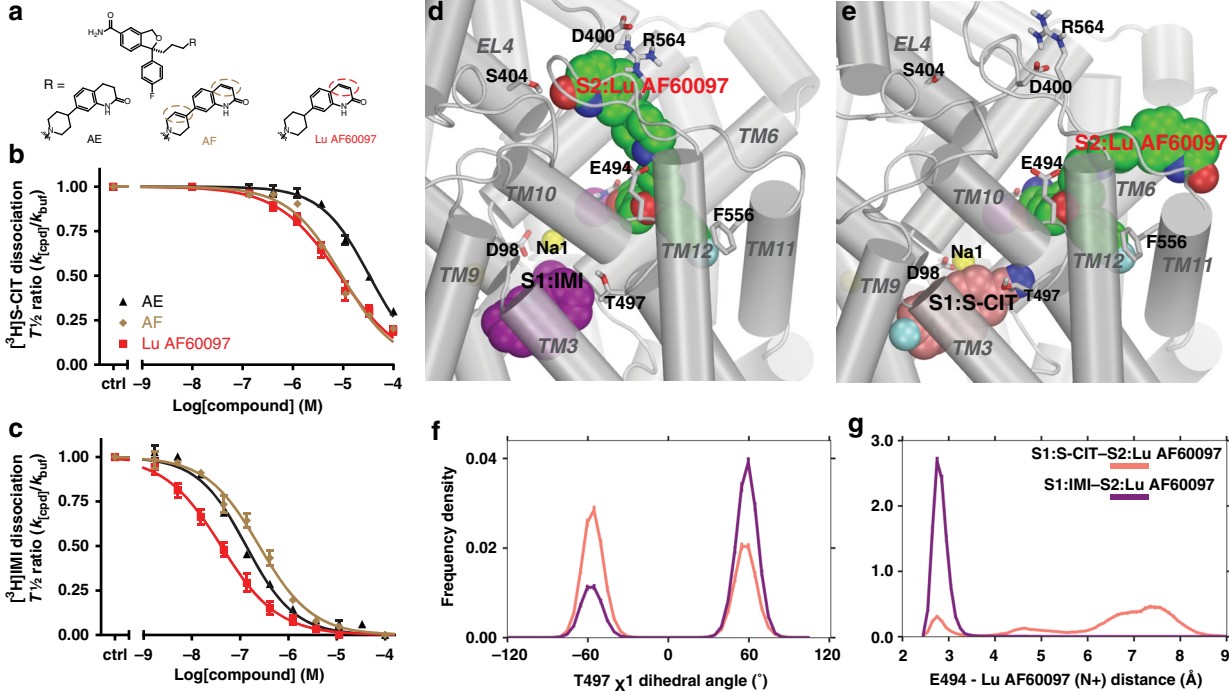

**Fig. 3 Identification of high-affinity allosteric inhibitors for SERT. a** Chemical structure of three tested compounds, all based on S-CIT template with an amide instead of the cyano in S-CIT. They differ in having either none (AE), one (Lu AF60097) or two (AF) double bonds (stippled circles) in their N-substituent. **b** The allosteric potency of the compounds inhibiting the dissociation of [³H]S-CIT is within the micromolar range: Lu AF60097 (red), $IC_{50}$ = 6.50 [5.07; 8.31] μM; AF (brown), $IC_{50}$ = 10.4 [8.45; 12.9] μM; and AE (black), $IC_{50}$ = 33.8 [30.7; 37.3] μM. Data are mean [S.E. interval], $n$ = 3-5. **c** The allosteric potency increases to nanomolar concentrations when assessed by inhibition of [³H]IMI dissociation. $IC_{50}$ values (in nM) for Lu AF60097, AF and AE are 31.4 [25.2; 39.1], 192 [173; 215], and 119 [103; 138], respectively (mean [S.E. interval], $n$ = 3-6. **d** A zoomed-in view of S1:IMI (in purple)/S2: Lu AF60097 (in green), and **e** S1:S-CIT (in salmon)/S2:Lu AF60097 (in green) conditions. **f** Distribution of the Thr497 $\chi_1$ rotamer for S1:S-CIT/S2: Lu AF60097 (salmon) and S1:IMI/S2:Lu AF60097 (purple) conditions. **g** Distribution of the Glu494/S2:Lu AF60097 distance (minimum distance between the charged N of Lu AF60097 and the two carboxyl oxygens of Glu494) for S1:S-CIT/S2:Lu AF60097 (salmon) and S1:IMI/S2:Lu AF60097 (purple) conditions. Experiments in **b** and **c** are performed essentially as in Fig. 1 on membrane preparations from COS-7 cells transiently transfected with SERT WT. Data are shown as means ± SEM (error bars). Source data are provided as a Source Data file.

higher allosteric potency in the presence of S1:IMI must not result in steric crowdedness near the Thr497-Phe335 motif, while forming a favored interaction with Glu494. To further characterize the potential of the S2 site as a druggable allosteric site[28,31] with the ultimate goal of developing therapeutic agents against it, we screened a compound library of citalopram analogs and assessed their allosteric potency using both S1:[³H]S-CIT and S1:[³H]IMI as described above.

Strikingly, we found three S-CIT analogues possessing very potent (30–200 nM) inhibition of the S1:[³H]IMI dissociation, but low (6–30 μM) allosteric potency on inhibiting the S1:[³H]S-CIT dissociation (Fig. 3a–c, Table 1). Interestingly, these three compounds all have a carboxamide instead of the cyano group on the benzofuran moiety of the S-CIT scaffold. They differ, however, in the presence and position of double bonds in the bicyclic N-substituents (Fig. 3a). In particular, the allosteric potency of Lu AF60097 in the presence of S1:IMI was 31 nM (Table 1), which is more than a 150-fold increase compared with citalopram.

To understand the molecular mechanism of the selective high allosteric potency of Lu AF60097, we characterized and compared the binding modes of S2:Lu AF60097 in the presence of S1:S-CIT versus S1:IMI by MD simulations. We first docked Lu AF60097 unbiasedly into the extracellular vestibule of our equilibrated S1: IMI/S2:apo model, and identified three poses (denoted as pose "I", "II", and "core") to be further relaxed and evaluated by MD simulations (see "Methods" and Supplementary Table 1). We then carried out molecular mechanics/generalized Born surface

area (MM/GBSA) calculations to evaluate the binding energy of the equilibrated poses, and found that the pose "core", in which the S-CIT scaffold of Lu AF60097 adopts a similar orientation as S2:S-CIT in the S2 site, has the most favored energy (pose "I", −59.7 kcal/mol; pose "II", −73.7 kcal/mol; pose "core", −77.6 kcal/mol). Consistent with the predicted binding free energy, the S-CIT core of Lu AF60097 in pose "I" protrudes out of the EV and is not fully engaged with hSERT, resulting in drastically weaker binding (Supplementary Fig. 3a). Whereas pose "core" forms the ionic interaction with Glu494 and is in proximity to Lys490, pose "II" does not form interactions with either of these residues but a polar interaction with Asp328 (Supplementary Fig. 3b, c). Therefore, we chose the pose "core" for further analysis (Fig. 3d, e). Our mutagenesis results of Asp328, Lys490, and Glu494 indeed support pose "core" but not pose "II" (see below).

Our MD simulations show that the cyano-to-carboxamide substitution orients the benzofuran moiety of Lu AF60097 to move slightly away from the Thr497-Phe335 motif compared with S2:S-CIT: whereas the cyano group of S2:S-CIT points to a polar cavity under Gln332, the carbamoyl group of Lu AF60097 forms a H-bond to the sidechain of Gln332 (Supplementary Fig. 4). Such a rearrangement allows the sidechain of Thr497 to be in the preferred *gauche* + $\chi_1$ rotamer in the presence S1:IMI, while S2:Lu AF60097 forms a salt bridge with Glu494 through its charged N (Fig. 3f, g). In addition, the quinolinone moiety of Lu AF60097 protrudes into a sub-pocket near the tip of the extracellular loop 4b (EL4b) with the 2-oxo modification forming

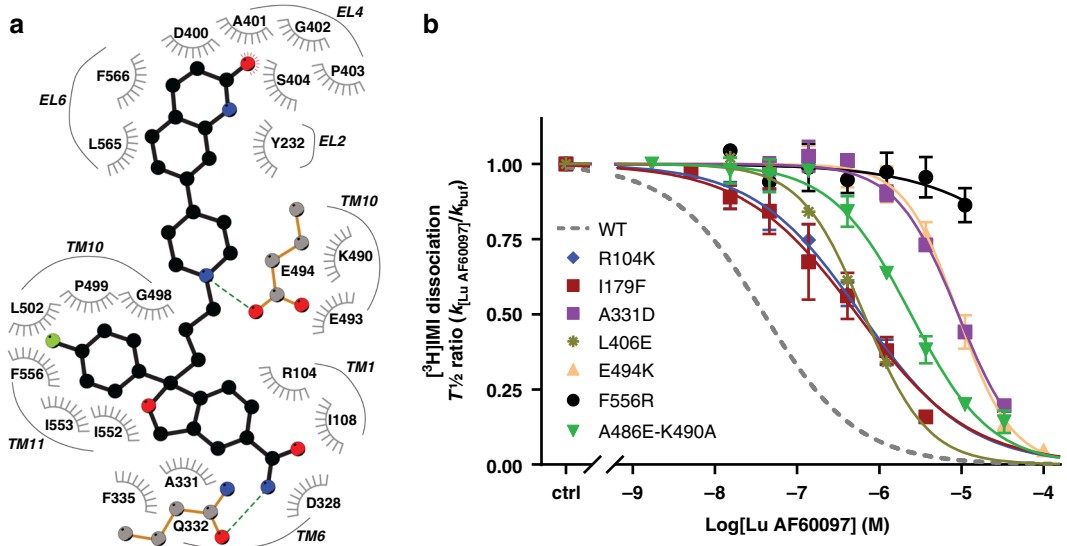

**Fig. 4 Evidence for binding of Lu AF60097 to the extracellular vestibule in SERT. a** A two-dimensional ligand interaction diagram[50] showing the residues interacting with Lu AF60097 that have >50% interaction frequencies (see Supplementary Table 3). **b** Effect of SERT point mutations on Lu AF60097 allosteric potency measured as inhibition of [³H]IMI dissociation (dotted line represents effect on SERT WT for comparison). The mutations are proposed to be located to the S2 site. The mutations cause a 14- to 293-fold decrease in S2:Lu AF60097 affinity (Table 2), with F556R not possessing any measurable affinity (IC$_{50}$ ≥ 10,000 nM). Allosteric potency on SERT WT is shown as dotted line. Data are shown as mean ± SEM (error bars), $n = 3$–6. Source data are provided as a Source Data file.

a hydrogen bond (H-bond) with Ser404 in EL4b (Fig. 3d and Supplementary Movie 1 and 2). Thus, the stability of this pose is likely associated with the structural integrity of EL4b and nearby interacting segments. Accordingly, mutations in EL4b and in neighboring TM3 are expected to destabilize this pose and reduce the binding affinity of S2:Lu AF60097 (see below).

Interestingly, when similar MD simulations were carried out for a S1:S-CIT/S2:Lu AF60097 model (Fig. 3e), in which the S-CIT scaffold of Lu AF60097 is aligned to S2:S-CIT as in the pose "core", S2:Lu AF60097 could not form the salt bridge with Glu494 through its charged N (Fig. 3g), while its quinolinone moiety could not get into the sub-pocket near EL4b. These divergent poses of S-CIT scaffolds of S2:Lu AF60097 versus S2:S-CIT are likely associated with the replacement of cyano group with a carboxamide in the benzofuran moiety. Consequently, the ionic interaction of S2:Lu AF60097 to Glu494 is much weaker than that of S2:CIT in the presence of S1:S-CIT (Figs. 2h and 3g). Moreover, by comparing the MD frames from both the S1:IMI/S2:Lu AF60097 and S1:S-CIT/S2:Lu AF60097 conditions when the quinolone moiety of Lu AF60097 did not protrude into the sub-pocket near EL4b, we found that the moiety is in different orientations and dynamics in the EV (Supplementary Fig. 5). Together, our computational results indicate that the allosteric interactions between the different S1:ligands and S2:Lu AF60097 result in markedly different poses of S2:Lu AF60097, which could account for the observed 200-fold loss in IC$_{50}$ in the presence of S1:[³H]S-CIT compared with that in the presence of S1:[³H]IMI (Table 1).

**The S2:Lu AF60097 binding mode is validated by S2 mutants.** From the analysis of equilibrated binding mode of S2:Lu AF60097 in the presence of S1:IMI, we identified residues from TMs 1, 6, 10, 11, EL2, and EL4 that directly interact with Lu AF60097 (Fig. 4a and Supplementary Table 3). To validate this prediction experimentally, we mutated individually or in combination a set of representative residues in TMs 1 (R104K), 3 (I179F), 6 (D328N and A331D), 10 (A486E-K490A and E494K/Q), 11 (F556R/L),

and EL4 (L406E). All the SERT mutants are able to bind and transport [³H]5-HT when expressed in COS7 cells (Table 2).

In the [³H]IMI dissociation assay, we found in agreement with the MD simulations that most mutations decreased the allosteric potency of Lu AF60097 more than tenfold (Fig. 4b, Table 2). The F556R mutant caused the most significant change with a complete ablation of the allosteric potency within the concentration range of the applied Lu AF60097. According to our MD simulations, Phe556 has a stable aromatic interaction with the fluorophenyl ring of the S-CIT scaffold in Lu AF60097, and therefore the F556R mutation would expectedly cause a drastic change in the IC$_{50}$. Ala331 interacts with both of the aromatic rings in the S-CIT scaffold of Lu AF60097 in the MD simulations, and the substitution with the negatively charged Asp residue does indeed cause a ~300-fold decrease in allosteric potency, suggesting a critical position of this residue in the binding pocket for S2:Lu AF60097. In addition, mutations L406E in EL4b and I179F in neighboring TM3 are expected to disrupt the structural integrity of EL4b, a segment predicted in accommodating the quinolinone moiety of Lu AF60097. Indeed, these mutations resulted in 21- and 14-fold decrease in the allosteric potency of S2: Lu AF60097, respectively. Taken together, most of the mutations based on the predicted pose "core" from our MD simulations have detrimental effects on Lu AF60097 binding in vitro, suggesting that the compound does bind to the predicted S2 site. Of note, E494Q only resulted in a ~5-fold decrease in allosteric potency. This is in contrast to the S1:S-CIT/S2:S-CIT condition in which the mutation caused a ~12-fold decrease. The difference is likely due to a reduced contribution from the salt bridge to binding affinity for the larger Lu AF60097 (39 heavy atoms, compared 24 of S-CIT). In our equilibrated S1:IMI/S2:Lu AF60097 model from the MD simulations, Lu AF60097 in pose "core" interacts with the backbone but not the sidechain of Asp328. Thus, the only minor (~2-fold) decrease in allosteric potency caused by the D328N mutation was expected. This result also argues that Lu AF60097 is less likely to be in pose "II", which forms a polar interaction with the sidechain of Asp328 (Supplementary Fig. 3b).

**Table 2 Effect of Lu AF60097 in inhibiting [³H]IMI dissociation from SERT WT and mutants.**

| SERT construct | $V_{MAX}$ [³H]5-HT (fmol/min/10⁵c) | $K_M$ (5-HT) (nM) | n | Lu AF60097 allosteric potency (nM) | n | ΔAP (mut/WT) |
|---|---|---|---|---|---|---|
| SERT WT | 4540 ± 740 | 520 [460;600] | 21 | 31 [25; 39] | 6 | – |
| R104K (TM1) | 480 ± 80 | 1100 [1000;1200] | 3 | 630 [460; 850] | 4 | 19 |
| I179F (TM3) | 1030 ± 430 | 2000 [1600;2400] | 3 | 440 [270; 710] | 5 | 14 |
| D328N (TM6) | 11300 ± 2200 | 1400 [970;1900] | 6 | 54 [45; 65] | 4 | 1.7 |
| A331D (TM6) | 5073 ± 418 | 290 [250; 330] | 3 | 9100 [9000; 9200] | 3 | 293 |
| L406E (EL4) | 114 ± 38 | 300 [280;320] | 5 | 650 [610; 700] | 5 | 21 |
| A486E (TM10) | 800 ± 140 | 320 [250;400] | 6 | 470 [390; 560] | 5 | 15 |
| A486E-K490A | 410 ± 140 | 190 [83;420] | 2 | 2400 [2100; 2700] | 3 | 77 |
| E494K (TM10) | 4700 ± 850 | 1400 [1200;1700] | 4 | 8900 [7500; 10500] | 3 | 286 |
| E494Q TM10) | 17100 ± 400 | 3900 [1900; 7700] | 2 | 170 [150; 180] | 3 | 5 |
| F556L (TM11) | 8300 ± 2000 | 1300 [1200; 1500] | 4 | 330 [310; 340] | 3 | 10 |
| F556R (TM11) | 9400 ± 1680 | 920 [700; 1200] | 7 | >10000 | 3 | >300 |

Characterization of [³H]5-HT transport and binding capacities for SERT WT and the investigated mutants (domain location in parentheses). 5-HT transport is performed on intact COS7 cells transiently expressing the indicated SERT construct. $V_{MAX}$ and $K_M$ for 5-HT is calculated based on the IC₅₀ and catalytic activity as described in Methods. Allosteric potencies are the IC₅₀ from non-linear regression analysis of the change in dissociation rate constant of [³H]IMI relative to no Lu AF60097 present ($k_{[Lu\ AF60097]}/k_{buffer}$) as a function of the added concentration (Log[Lu AF60097]). Allosteric potency experiments are performed on membrane preparations of COS-7 cells expressing SERT WT. Data are shown as mean and either ± SE (for $V_{MAX}$) or [SE interval] and are calculated from pIC50 and the SE interval from pIC50 ± SE. Note that value for SERT WT is the same data set as shown in Table 1. ΔAP (mut/WT), change in allosteric potency for mutants relative to SERT WT.

**Competitive and non-competitive inhibition of 5-HT transport.** Because of the low potency of S2:Lu AF60097 in inhibiting the dissociation of S1:[³H]S-CIT, we reasoned that it was possible to assess the affinity of Lu AF60097 at the S1 site using [³H]S-CIT equilibrium binding. The results of such experiments showed that the IC₅₀ of Lu AF60097 was 265 [196; 358] nM (mean [S.E. interval], n = 6, Table 1, Supplementary Fig. 6). Whereas we cannot rule out that [³H]S-CIT binding is partially modulated by the allosteric interaction, the results suggest that the S2 (in the presence of S1:IMI) over S1 selectivity for Lu AF60097 is at least eightfold. For S-CIT, the selectivity is ~1000-fold in favor of the S1 site. In comparison, when we performed a binding experiment under similar conditions but using [³H]IMI as the radioligand, [³H]IMI was not displaced by Lu AF60097, possibly because Lu AF60097 has a high S2 affinity in the presence of S1:IMI and locks [³H]IMI in the S1 site (Supplementary Fig. 6). This finding further substantiates the ligand-dependent differences in the allosteric interaction between the S1-and S2-bound ligands.

Next, we investigated the capability of Lu AF60097 to inhibit the transport of [³H]5-HT by hSERT expressed in COS-7 cells (Fig. 5). Lu AF60097 inhibited [³H]5-HT uptake with an IC₅₀ of 207 [202; 214] nM (mean [S.E. interval], n = 5, Fig. 5a). This IC₅₀ is different from its measured allosteric potency but similar to the equilibrium binding affinity for Lu AF60097 when displacing [³H]S-CIT, suggesting a S1 component of Lu AF60097's inhibition of the uptake. To further substantiate the possibility of a S1-binding component, we also performed [³H]5-HT uptake inhibition in the F556R mutant, which only showed very minimal allosteric binding by Lu AF60097 (Fig. 4). Indeed, the uptake inhibition by Lu AF60097 in this mutant was not different from WT ($K_i$ = 220 [190; 240] and 260 [210; 310] nM, n = 15 and 3, for WT and F556R, respectively, Fig. 5a).

To examine whether the inhibition of 5-HT uptake is due to a blockade by a competitive or a non-competitive action by Lu AF60097, we performed [³H]5-HT saturation uptake with increasing concentrations of Lu AF60097 (Fig. 5b, Supplementary Table 4). The results show that Lu AF60097 inhibits 5-HT uptake mainly by changing the $K_M$ of [³H]5-HT transport in the low concentrations, indicative of a competitive action. In the high concentrations Lu AF60097 also reduce the maximal uptake velocity ($V_{MAX}$), suggesting a combined competitive and non-competitive mechanism. Thus, together with the [³H]S-CIT equilibrium binding results, we found that Lu AF60097 possesses a S1-binding-based competitive component as well, when the S1 site is not occupied by IMI.

In our prolonged MD simulations of the potential binding pose of Lu AF60097 in the S1 site, we found that its S-CIT scaffold adopts a similar pose as that of the S1-bound S-CIT, while the quinolinone moiety protrudes out of the S1 site. However, its positively charged tertiary amine moiety cannot form any ionic interaction with either the sidechain carboxyl group of Asp98 or the backbone carbonyl group of Tyr95 (Supplementary Fig. 7)[38]. This less favored binding mode of Lu AF60097 is consistent with its significantly reduced affinity at S1 compared with S1:S-CIT (Table 1).

**Lu AF60097 and imipramine can block SERT synergistically.** Since the Lu AF60097 affinity is markedly increased in the presence of S1:IMI, we predicted that they would inhibit 5-HT uptake synergistically, i.e., the inhibitory effect of applying them together would be more potent than combining the effects of applying them individually. Thus, we studied the inhibition of [³H]5-HT uptake by low concentrations of either IMI or Lu AF60097 alone or in combination. As shown in Fig. 5c, IMI (4 nM) or Lu AF60097 (27 nM) alone only resulted in a modest decrease in [³H]5-HT uptake (10.7 ± 0.8% and 5.0 ± 1.8% inhibition, respectively, relative to control, mean ± S.E., n = 5–7). In contrast, the two compounds together acted synergistically and caused a significant 36.2 ± 0.3% decrease of the 5-HT uptake (mean ± S.E., n = 5). The results support that the binding of one ligand facilitates the binding of the other ligand.

**Hippocampal 5-HT levels are increased by Lu AF60097.** To investigate whether Lu AF60097 administration has any effect on 5-HT homeostasis in an in vivo setting, we performed microdialysis in rat hippocampus with local administration of Lu AF60097. Two microdialysis probes were inserted, one in each hemisphere, into rat hippocampus. In close vicinity of each probe, an injection needle was placed and 5-HT levels were measured through the microdialysis probe on the freely moving rats. When 5-HT levels were stabilized, we injected 1 μl of 250 nM Lu AF60097 in one hemisphere while saline (artificial cerebrospinal fluid, aCSF) was injected in the other for comparison. Lu AF60097 increased 5-HT levels reaching significance after 80 min and reached to about 6-fold above saline after 2 h (Fig. 6a). The

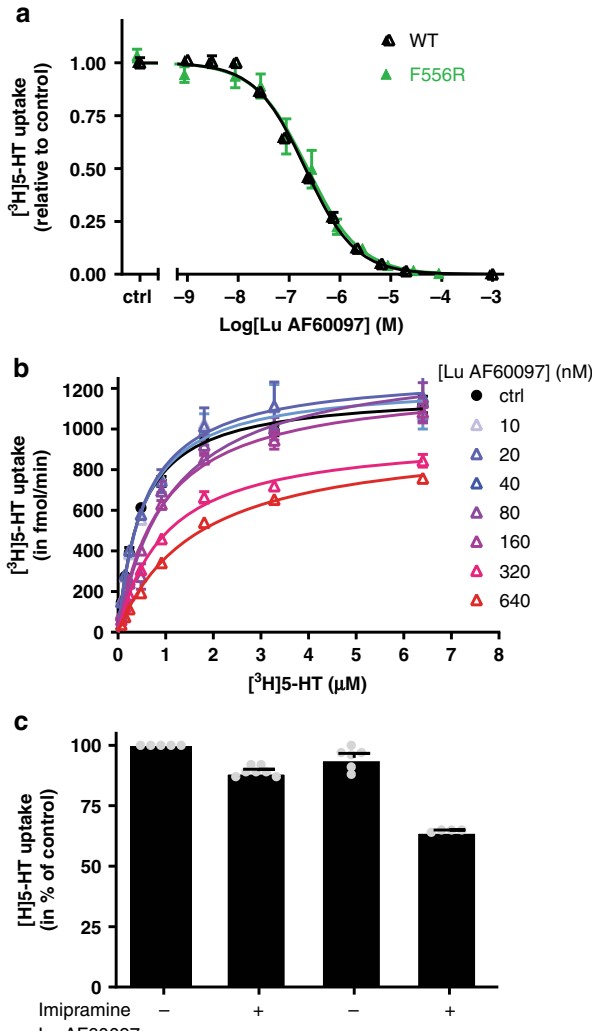

**Fig. 5 Pharmacologic properties of Lu AF60097 binding to SERT. a** Inhibition of [³H]5-HT uptake by Lu AF60097 in SERT WT and F556R. Intact COS-7 cells were pre-incubated with Lu AF60097 for 15 min before [³H]5-HT uptake was initiated. **b** Lu AF60097 inhibits [³H]5-HT uptake with a mixture of competitive and non-competitive mechanisms. Saturation uptake experiments for [³H]5-HT transport as a function of increasing Lu AF60097 concentrations (0–640 nM). The $V_{MAX}$ decreases with increasing Lu AF60097 (from 1180 ± 20 fmol/min (no Lu AF60097) to 920 ± 20 fmol/min (640 nM Lu AF60097). In addition, the $K_M$-value for 5-HT increases with increasing concentration of Lu AF60097 ($K_M$ = 484 nM and 1650 nM for buffer only and 640 nM Lu AF60097, respectively (Supplementary Table 4). **c** Application of Lu AF60097 and imipramine together inhibits [³H]5-HT uptake synergistically. Imipramine (4 nM) and Lu AF60097 (27 nM) results in a minor inhibition of [³H]5-HT uptake (to 89.3 ± 0.8 and 94 ± 2% of control, respectively). In contrast, adding the same concentration of the two compounds together results in a significantly higher decrease of [³H]5-HT uptake (64.8 ± 0.3%). ****$p$ < 0.001 relative to addition of either imipramine or Lu AF60097 alone (one-way ANOVA with Tukey's multiple comparisons test). All experiments are performed on intact COS-7 cells transiently transfected with SERT WT. Data are shown as means ± SEM (error bars) of at least three experiments performed in triplicates. Source data are provided as a Source Data file.

results suggest that Lu AF60097 is capable of targeting and blocking SERT function in vivo at a concentration no higher than 250 nM. We further assessed whether it was possible to mimic the synergistic effect on 5-HT levels we observed on cell lines when

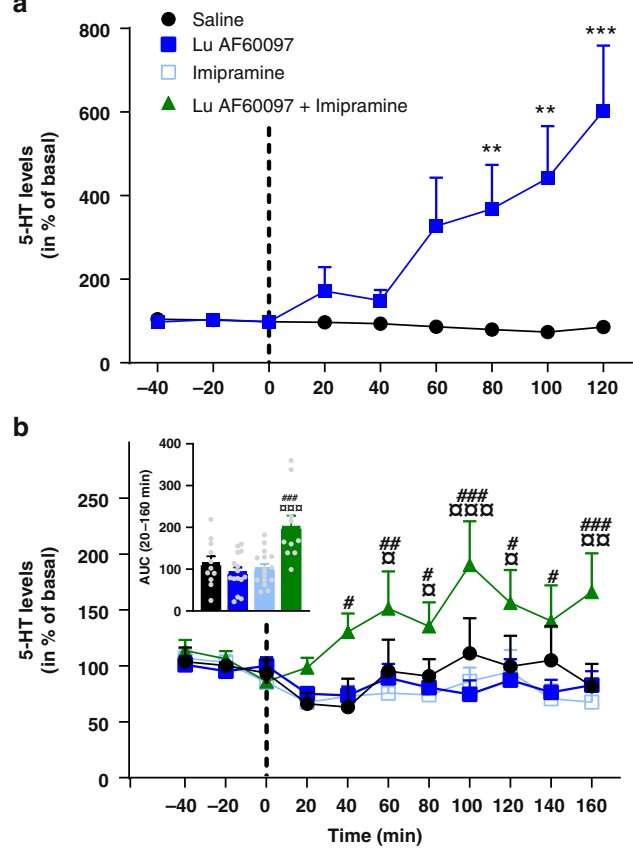

**Fig. 6 Effects of Lu AF60097 and imipramine on 5-HT levels in rat hippocampus. a** Changes in extracellular 5-HT levels after injection (dotted line) of 1 µl (250 nM) Lu AF60097 (squares) or saline (circle) in ventral hippocampus of freely moving rats. The 5-HT levels show a continuous increase at all time points after the 40 min post injection bin to a maximum of 602 ± 156% of basal level (data are means ± S.E.M., $n$ = 6). **b** Hippocampal perfusion of Lu AF60097 and imipramine synergistically increases extracellular 5-HT levels. Extracellular 5-HT levels were determined 40 min prior and 160 min after initiating (dotted line) a continuous perfusion of either saline (aCSF, circles), 1.9 µM Lu AF60097 (closed squares), 0.36 µM imipramine (open squares), 1.9 µM Lu AF60097 + 0.36 µM imipramine (triangles) in ventral hippocampus of freely moving rats. Inset: Area under the curve during perfusion (20–160 min) for the four conditions (matching colors to curves in main figure). The released 5-HT when perfused with both imipramine and Lu AF60097 is significantly increased (green bar). When perfused with either of the compounds alone (blue bars), the 5-HT levels are not different form saline (black bar). Data are means ± SEM (error bars), $n$ = 9–15. The extracellular concentrations of 5-HT in both (**a**) and (**b**) are assessed by microdialysis and expressed as the percentage of the basal levels in the three fractions collected prior to perfusion start (dotted line). **$P$ < 0.01, ***$P$ < 0.001 vs. saline; ¤$P$ < 0.05, ¤¤$P$ < 0.01, ¤¤¤$P$ < 0.001 vs. Lu AF60097; #$P$ < 0.05, ##$P$ < 0.01, ###$P$ < 0.01 vs. imipramine; Bonferroni post-hoc test after significant RM two-way ANOVA or one-way ANOVA. Source data are provided as a Source Data file.

co-administering IMI and Lu AF60097. To match the concentrations from the in vitro experiment, we first performed an in vitro recovery experiment to determine the fraction of compounds, which would perfuse through the dialysis probe. We found that 18.6 ± 1.1% and 3.22 ± 0.7 (means ± SEM, $n$ = 3) of IMI and Lu AF60097, respectively would cross the probe membrane. Based on these experiments, we found that the perfusion of either 0.36 µM IMI or 1.9 µM Lu AF60097 in the microdialysis

probe had no effect on hippocampal 5-HT levels relative to saline within the duration of the experiment (Fig. 6b). In contrast, when the two compounds were administered together, the 5-HT levels rose to significantly higher levels either 40 or 60 min after perfusion start, relative to IMI and Lu AF60097 alone, respectively. The increased 5-HT levels remained increased during the remaining of the 160 min test period. These data suggests that Lu AF60097 is also able to potentiate the effect of IMI on extracellular 5-HT levels in an in vivo setting.

## Discussion

The presence of an allosteric site in SERT has been known for more than three decades[39]. Structurally diverse compounds such as sertraline[17], paroxetine[19], clomipramine[21], and citalopram[21] have been shown to possess allosteric activity as they can impair dissociation of a pre-bound high affinity radioligand to the transporter. However, the allosteric potencies of these compounds are all in the micromolar range, while they bind to the orthosteric site with low-nanomolar affinity. Accordingly, it has not been possible to isolate their specific allosteric impact on SERT function. Mutagenesis studies[21] and x-ray crystallography[11] have located the allosteric site to the EV of the transporter. In addition to SERT, allosteric sites at similar locations have been found in NET[18] and LeuT[26]. Further, it has been proposed that S2 occupancy by a substrate in LeuT is required for substrate to be released from S1[24], while the substrate 5-HT has been shown to have an allosteric effect on [3H]IMI dissociation in SERT, though with low potency[39].

Here, we show at atomistic detail that the S1 and S2 binding sites in SERT are allosterically coupled to each other. By combining extensive (~125 μs) MD simulations of various conditions and site-directed mutagenesis, we show that the impact of ligand binding to the S1 site propagates through the Thr497-Phe335 motif to alter the configuration of the S2 site. In particular, the propensity of Glu494 in the S2 site to form a salt bridge with the S2-bound ligands is differentially affected by S1:S-CIT and S1:IMI. Consistent with this prediction, removal of the negative charge of Glu494 (E494Q) results in similar S2 affinities for S-CIT in the presence of either S1:IMI or S1:S-CIT, while the same impact on the S2 affinities of R-CIT were observed as well (Fig. 1). Moreover, mutating Thr497 to a residue with a smaller sidechain (T497A) improves S2:S-CIT binding in the presence of S1:[3H]IMI, to the same extent as when [3H]S-CIT is bound to S1. Together, the results suggest that the configuration of the Thr497-Phe335 motif is sensitive to the identity of the S1-bound ligand and plays a critical role in the allosteric communication between the S1 and S2 sites.

We further report on a compound with nanomolar affinity at the S2 site of hSERT. Lu AF60097 has a >100-fold gain in allosteric potency relative to any other reported S2-bound hSERT ligand[22]. Our MD simulations suggest that Lu AF60097 can stably bind in the EV with its S-CIT scaffold in a similar pose as that of S2:S-CIT revealed by the hSERT crystal structure[11]. Interestingly, the potency of Lu AF60097 in the S2 site shows a reversed trend compared with S2:S-CIT, i.e. higher allosteric potency with S1:[3H]IMI than with S1:[3H]S-CIT (Fig. 3b, c). Based on our simulation results, we propose that, compared with S2:S-CIT, the carboxamide substituent of Lu AF60097 alters the polar interactions near Gln332 and shifts the S-CIT scaffold slightly away from the Thr497-Phe335 motif thus relieving the steric crowdedness between the S1 and S2 sites in the presence of S1:IMI. In addition, our simulation results indicate that the selectively improved S2:Lu AF60097 affinity in the presence S1:IMI may also come from the specific binding of the quinolinone

substituent of Lu AF60097 in a sub-pocket near EL4, which is not formed in the presence of S1:S-CIT. Mutations of selected residues show detrimental effects on Lu AF60097 allosteric potency, supporting its predicted binding pose. Taken together, we conclude that, in the presence of S1:IMI, Lu AF60097 binds with high affinity to the EV of SERT.

In addition to high S2 affinity, high S2 specificity is also necessary to isolate the allosteric impact of a S2-bound ligand on SERT function. Thus, we investigated whether Lu AF60097 has any S1 binding component and found that it binds at S1 with an IC_{50} of ~ 265 nM (Supplementary Fig. 6). This value is ~9-fold higher than that of its allosteric potency and is promising for isolating the allosteric impact. Supported by the [3H]5-HT saturation uptake experiment (Fig. 5b) and the F556R mutation, which virtually eliminates S2:Lu AF60097 binding, but has no impact on its inhibition of the 5-HT uptake, we conclude that Lu AF60097 has a S1 binding component.

Whereas we present the first lead compound with high-affinity allosteric association to SERT in vitro, an immediate question is whether this translates into an in vivo setting. We showed that a small amount of Lu AF60097 (1 μl 250 nM) is able to elicit a marked increase of extracellular 5-HT levels in the microdialysis analysis, suggesting that the compound is also capable of targeting SERT in vivo. This we further substantiated by showing that co-administration of IMI and Lu AF60097 in vitro (Fig. 5c) and in vivo (Fig. 6b) does have a potent effect on the inhibition of 5-HT uptake, compared with administrating either of these two compounds alone in the same concentrations. This opens the door for further in vivo analysis of Lu AF60097 or similar next generation compounds to adequately assess these potentials. We propose that a clinical potential of the allosteric inhibitor property of Lu AF60097 lies in its potentiation of IMI binding. This might make it possible to lower the therapeutic dose of IMI by co-administering an allosteric binder such as Lu AF60097, preserving the positive effects of tricyclic antidepressants on major depressions while reducing the detrimental side effects such as cardiac arrhythmias and -arrest[40] and interferences with autonomic control[41]. Indeed, the results herein open the possibility of performing additional in vivo experiments with SERT allosteric inhibitors in combination with current effective orthosteric inhibitors, to probe for improved therapeutic effects using behavioral paradigms for depression or anxiety.

## Methods

**Site directed mutagenesis**. The human SERT was cloned into the pUbi1z vector using the NotI and XbaI. Mutations herein were generated using the QuickChange method (adapted from Stratagene, La Jolla, CA) or ordered through GeneArt. All mutations were confirmed by DNA sequencing. The used primers were

L406E, CGCAGGTCCCAGCCTCGAGTTCATCACGTATGCAG
A486E, CTTTTTGGAGGGGAGTACGTGGTGAAG
E494K, GAAGCTGCTGGAGAAGTACGCCACGGGG
F556L, CATTTGCAGTTTACTCATGAGCCCGCCAC
F556R, CATCATTTGCAGTAGACTGATGAGCCCG

Only the sense primers are shown. The complimentary antisense primers were also used.

**Transfection and Membrane preparation**. Membranes were prepared from COS-7 cells (a generous gift form Prof. U Gether, U. of Copenhagen, DK), after they were transient transfected with hSERT WT or mutants using the Lipo2000 transfection protocol (Invitrogen): 2 μg SERT plasmid and 6 μL Lipofectamine were mixed each with 250 μL Opti-Mem^R(1×) and incubated for 20 min for micelle formation. The mixture was added to 10 ml DMEM 1885 medium containing 4 million COS-7 cells. After 5 h, 25 ml DMEM1885 with antibiotic was added. After 48 h. the cells were harvested with PBS + 5 mM EDTA, washed, and lysed with 1 ultrasound burst (Branson Sonifier with microtip) in membrane buffer (120 mM NaCl, 5 mM KCl, 1.2 mM MgSO_4, 1.2 mM CaCl_2, 25 mM HEPES, pH 7.4). Membranes were pelleted at $4900 \times g$ and resuspended in membrane buffer containing 0.3 M sucrose and stored at −80 °C until used.

[3H]IMI and [3H]S-CIT dissociation rate assay 25 nM [3H]IMI (40–50 Ci/mmol) or [3H]S-CIT (85 Ci/mmol) were added to membranes incubated for 30 min at 0 °C. Dissociation was initiated by 12× dilution of membrane aliquots with membrane buffer containing 0.1 μM paroxetine and the indicated concentrations of allosteric inhibitor. Note: 0.1 μM paroxetine has no allosteric effect on the dissociation of [3H]IMI and [3H]S-CIT. The dissociation was assessed at seven time points (5–10–15–20–30–50–70 min) and stopped by rapid filtration of the samples through GF/B filters using a Tomtec cell harvester and washed for 20 s with ice-cold 0.2 M NaCl. Experiments were performed in a water bath at a temperature where t½ for control dissociation (i.e. dissociation without drug) were set to ~15 min. All experiments were performed in at least three independent experiments.

[3H]S-CIT and [3H]IMI competitive binding experiments were performed on membrane preparations of COS-7 cells transiently transfected with SERT WT. Membrane aliquots were mixed with membrane buffer containing 5–10 nM of either [3H]S-CIT or [3H]IMI and Lu AF60097 in the indicated concentrations to a final volume of 300 μL. The binding experiments were incubated 1 h at room temperature and subsequently filtered, washed and counted as described under '[3H]IMI and [3H]S-CIT dissociation rate assay'. Non-specific binding was determined by adding 5 μM paroxetine.

[3H]5-HT uptake experiments were performed using 5-[1,2-3H] hydroxytryptamine (25–30 Ci/mmol). COS-7 cells, transfected with SERT WT or SERT mutants were seeded in 24-well dishes ($10^5$ cells/well) coated with poly-ornithine. The seeded cell number were adjusted to achieve an uptake level of maximally 10% of total added [3H]5-HT. The uptake assays were carried out 2 days after transfection. Prior to the experiments the cells were washed once in 800 μL uptake buffer (120 mM NaCl, 5 mM KCl, 1.2 mM $MgSO_4$, 1.2 mM $CaCl_2$, 10 mM glucose, 25 mM HEPES, pH 7.4) at room temperature. Drugs tested for inhibition of uptake were added to cells in indicated concentrations 30 min prior to addition of 0.3 μCi [3H]5-HT. After incubating 3 min (WT) or 5 min (mutants) the cells were washed twice with 500 μL ice-cold uptake buffer, lysed in 250 μL of 1% SDS and left for 1 h at 37 °C. All samples were transferred to 24-well counting plates, and 500 μL Opti-phase Hi Safe 3 scintillation fluid was added followed by counting in a Wallac Tri-Lux beta-scintillation counter. Nonspecific uptake was determined in the presence of 1 μM paroxetine. All determinations were performed in triplicate.

**Molecular docking.** We used the crystal structures of hSERT bound with S-citalopram in the S1 site only (PDB ID 5I71), and bound in both S1 and S2 sites (PDB ID 5I73) in ts3 construct as the starting points for our modeling studies. The binding site ions missing in the crystal structures were added. For the WT hSERT models, the three thermostabilizing mutations that were introduced in the ts3 construct were mutated back to the WT residues.

Imipramine was docked into the S1 site using the induced-fit docking (IFD) protocol[42] implemented in the Schrodinger suite (release 2016-4). The best-scoring pose was selected, which is consistent with previously deduced imipramine binding pose[43].

Lu AF60097 was docked into the S2 site using the IFD protocol. The RMSD values of the ensemble of IFD poses were calculated and clustered using the centroid linkage method as implemented in the Conformer Cluster module of Maestro software (release 2016-4, Schrodinger Inc., New York, NY). Three largest clusters of Lu AF60097 poses were: pose I, where dihydroisobenzofuran moiety is close to Phe556 and fluorophenyl is close to TM1b and TM6; pose II, where fluorophenyl is close to Phe556 and dihydroisobenzofuran moiety is close to TM1b and TM6; and pose "core", where the benzodioxo and fluorophenyl moieties of the S-CIT scaffold adopt a similar orientation as S2:S-CIT (PDB ID 5I73) and the quinolinone moiety of Lu AF60097 protrudes toward extracellular milieu. These three poses were selected as initial starting points for further relaxation by MD simulations (Supplementary Table 1) and evaluation by MM/GBSA for their binding energy (see below).

**MM/GBSA analysis.** The Molecular mechanics/generalized Born surface area calculations (MM/GBSA) analysis of the S1:IMI/S2:Lu AF60097 binding poses was carried out on the last 300 ns of each trajectory using the thermal_mmgbsa.py script from Schrodinger suite (release 2017-2), which calculates Prime MM/GBSA (version 4.8) for every frame in a trajectory using OPLS3 force field with VSGB2.1 solvation model[44]. The values reported in the results section are averages for all trajectories of each of the three S1:IMI/S2:Lu AF60097 poses.

**MD simulations.** hSERT models were placed into explicit 1-palmitoyl-2-oleoyl-sn-glycero-3-phosphocholine lipid bilayer (POPC) using the orientation of the 5I73 structure from the Orientation of Proteins in Membranes database[45]. Simple point charge (SPC) water model[46] was used to solvate the system, charges were neutralized, and 0.15 M NaCl was added. The total system size was ~135,000 atoms. Desmond MD systems (D. E. Shaw Research, New York, NY) with OPLS3 force field[47] were used for the MD simulations. The system was initially minimized and equilibrated with restraints on the ligand heavy atoms and protein backbone atoms, followed by production runs at 310 K with all atoms unrestrained. The NPγT ensemble was used with constant temperature (310 K) maintained with Langevin dynamics, 1 atm constant pressure achieved with the hybrid Nose-Hoover Langevin piston method[48] on an anisotropic flexible periodic cell, and a constant surface tension (x-y plane). Overall, 74 trajectories of with a total simulation time of 125.28 μs were collected (Supplementary Table 1).

**Conformational analysis.** Distances and dihedral angles were calculated with MDTraj (version 1.7.2[49]), in combination with in-house Python scripts. Data sets for conformational analyses were assembled as follows. We first combined data from individual trajectories into a common pool for each simulated condition. Then, for the histograms in Figs. 2g, h and 3f, g, we extracted 500 bootstrapped samples of 5000 random frames each for a given simulated condition and plotted averages and standard deviations of frequency distributions for those 500 samples. For the ligand pairwise RMSD calculations (Fig. 2i) we carried out ten bootstrap samplings, and extracted 500 frames for each sampling for each condition. For each of the 500-frame bootstraps, all the frames are aligned pairwise for the RMSD calculations, yielding a 500 × 500 matrix. The averages and standard deviations of the ten bootstrapped samples are reported. For Supplementary Fig. 5, 50,000 random frames were extracted from the S1:S-CIT/S2:Lu AF60097 and S1:IMI/S2:Lu AF60097 conditions (for the latter, we selected and used the frames that have the distance between the 2-oxo modification of the quinolinone moiety of AF60097 and the side chain oxygen of Ser404 greater than 4.5 Å).

**Microdialysis experiments.** Sprague Dawley rats weighing 300–400 g is anesthetized with Hypnorm/Midazolam (2 ml/kg) and intracerebral guide cannulas are stereotaxically implanted into the brain, aiming to position the guide cannula tip in the ventral hippocampus (co-ordinates: −5.6 mm posterior to bregma, lateral −4.8 mm, −4.0 mm ventral to dura) according to Paxinos and Watson. Anchor screws and dental cement are used for fixation of the guide cannulas. The body temperature of the animals is maintained at 37 °C using a homoeothermic blanket. Rats recovered from surgery for 2–3 days, single-housed.

On the day of the experiment a microdialysis probe (CMA/12, 0.5 mm diameter, 3 mm membrane length, non-metal) is inserted through the guide cannula. The probes are connected via a dual swivel to a microinjection pump. Perfusion of the microdialysis probe with filtered Ringer solution (145 mm NaCl, 3 mM KCl 1 mM $MgCl_2$, 1.2 mM $CaCl_2$) begin shortly before insertion of the probe into the brain and continued for the duration of the experiment at a constant flow rate. After stabilization, the experiments are initiated. The experimental design consisted of collection of brain dialysate samples in 20 min fractions. Prior to the first sample collection the probes had been perfused for 180 min. A total of 12 fractions were sampled (four basal fractions and eight fractions after perfusion start) and the dialysate 5-HT content was analyzed using HPLC detection. Perfusion solutions contained Lu AF60097, imipramine, a combination of the two, or vehicle (artificial cerebrospinal fluid (aCSF)). After the experiments, the animals are euthanized.

The concentration of 5-HT in the dialysates was determined by means of HPLC with electrochemical detection. The monoamines were separated by reverse phase liquid chromatography (ODS 160 × 3.0 mm column) and analyzed using a mobile phase consisting of 150 mM $NaH_2PO_4$, 4.8 mM citric acid monohydrate, 3 mM dodecyl sulfate, 50 μM EDTA, 8 mM NaCl, 11.3% methanol and 16.7% acetonitrile (pH 5.6); flow rate of 0.4 ml/min. Electrochemical detection was accomplished using a coulometric detector and a SenCell (Antec); potential set at E1 = 500 mV (Coulochem III, ESA).

All experiments involving research animals were performed in accordance with guidelines from the Danish Animal Experimentation Inspectorate and approved by the local ethical committee.

**Data calculation.** The allosteric potency was calculated as previously described[21]. The dissociation rate constants ($k_{[drug]}$) at indicated unlabeled ligand concentrations were calculated and expressed relative to the dissociation rate constant without the presence of unlabeled ligand ($k_{buffer}$). The allosteric potency was determined as the drug concentration that impairs the dissociation rate by 50% compared with dissociation in buffer. $IC_{50}$ values were calculated from concentration effect curves of normalized dissociation ratio ($k_{[drug]}/k_{buffer}$) versus log [drug] and are shown as mean values calculated from means of pIC50 and the SE interval from the pIC50 ± S.E. All data were analyzed by linear or nonlinear regression analysis using Prism 7.0 (GraphPad Software Inc., San Diego, CA).

**Reporting summary.** Further information on research design is available in the Nature Research Reporting Summary linked to this article.

## Data avaliability

Data supporting the findings of this manuscript are available from the corresponding authors upon reasonable request. A reporting summary for this article is available as a Supplementary Information file.

The source data underlying Figs. 1b–e, 3b–c, 4b, 5a, 6 and Supplementary Fig. 6 are provided as a Source Data File.

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

## Acknowledgements

We thank Lone Rosenquist, Bente Bennicke, Henrik Pedersen, Krestian Larsen, Nina Guldhammer and Sascha Bull for excellent technical assistance, Klaus Bøgesø, Jonathan A. Javitch, Satinder K. Singh, and Kristian Strømgaard for insightful discussions. This research was supported in part by the Intramural Research Program of the NIH, NIDA.

## Author contributions

L.S., U.G., P.P., B.B.A., and C.J.L. conceptualized the ideas. P.P., A.M.A., B.B.A., L.S., U.G., and C.J.L. designed the experiments with support from P.W. and G.S. P.P. performed the experiments and data analysis together with P.W. and C.J.L. B.B.A. was responsible

for the synthesis of compounds. A.M.A. and L.S. designed and carried out the computational modeling, simulations and analysis. All authors were involved in data interpretation. C.J.L. and L.S. prepared the initial manuscript with contribution from all the authors.

## Competing interests

The authors declare no competing interests.
