## [Peer Review File · Nature Communications]

Reviewers' Comments:

Reviewer #1:

Remarks to the Author:

This manuscript focuses on the ability of some ligands for the serotonin transporter (SERT) to bind in the external vestibule of the transporter in addition to their primary site of binding at the locus where substrate (serotonin) binds. A second binding site was originally invoked to explain why high concentrations of substrate and some inhibitors could slow down dissociation of bound radioligands. The appearance of a second molecule of bound citalopram in the extracellular vestibule of SERT X-ray structures provided further encouragement for those attracted to the idea that transporters in this family had evolved an additional site for substrate and inhibitors in the service of some ill-defined physiological function. The alternative explanation for ligands to bind in this location is that it contains a mixture of polar and non-polar residues that interact to seal the external vestibule (EV) closed and that ligands present at high concentrations just stick there.

Irrespective of why this site exists, it is a shiny object that attracts attention, and is the subject of this manuscript. The authors have examined the interaction between a molecule of citalopram bound in this EV site and citalopram or imipramine bound in the high-affinity site. The two ligands are separated by about 6Å in the structure, and several side chains interact with both ligands. The first half of the work reported here concerns the interaction between the two molecules. The authors find that citalopram affinity in the EV site is higher when imipramine is bound at the substrate site than when citalopram is bound there. This interaction can be modulated by mutation of some residues whose side chains contribute to the regions common to the two binding sites. The observations are quite consistent with the structures and the computational modeling and mutagenesis support the authors' conclusions.

The remainder of the manuscript is devoted to the identification of an inhibitor from the Lundbeck collection of citalopram derivatives, which has higher affinity for the EV site than other inhibitors. However, this affinity, like that of citalopram, is higher when imipramine is bound at the substrate site. The authors model the derivative, Lu AF60097, into the EV site, and validate the binding pose with mutagenesis. They show that it binds both to the substrate site and the EV site and that the affinity to the EV site with imipramine in the substrate site is even higher than its affinity for the (empty) substrate site.

These results predict that Lu AF60097 should inhibit serotonin transport synergistically with imipramine and this is observed *in vitro*. However, the only *in vivo* experiment shown is with Lu AF60097 alone. It inhibits, but why wasn't the synergy with imipramine tested?

The purpose of all this effort, purportedly, is to take advantage of the EV site to develop therapeutic drugs that would enhance the potency of existing drugs. If this ability to enhance imipramine was not tested, it should have been. If it was, the results should be included.

The point of this manuscript seems to be the therapeutic potential of EV ligands. However, the proof of principle *in vivo* seems missing. The other point made in the beginning of the manuscript is that ligand binding at the two adjacent sites can influence each other. I don't see this as very surprising, and it may be a bit over-sold as in the statement that binding in the EV site "is associated with robust conformational rearrangements". Conformational rearrangements in transporters implies changes that convert open-out to open-in forms. In this instance, they are side chain movements. The documentation of these movements and their consequences seems quite solid, but the results are not surprising and do not, in my mind, represent a fundamental increase in our understanding of ligand binding in SERT.

Minor points:

The second sentence in the last paragraph on page 7 is unclear to me. It begins by describing the gauche rotamer of T497 when citalopram is bound at the substrate site and contrasts that to when imipramine is bound at the substrate site, which forces T497 into... the gauche rotamer?

Figure 2F is unclear. Here the conformation of the T497 side chain seems to change, unlike the text description.

Why does almost every observation have to be "striking"?

In the first sentence on page 9, it's not clear what a "more optimal" configuration would be. Please

describe what is meant here.

In the next paragraph, the sentence beginning "Interestingly," is overly complex. Try splitting into two smaller sentences.

The Y axis in Figure 5c is labeled Inhibition of uptake but looks more like uptake.

Reviewer #2:

Remarks to the Author:

This manuscript presents a very thorough characterization of a novel high affinity allosteric inhibitor of SERT. As SERT is the target of many psychoactive drugs and no compounds with high affinity for the S2 site have been previously reported, the potential significance of this work is high.

the authors investigate the basis of allosteric coupling between the S1 and S2 sites primarily using molecular dynamics simulations, although the conclusions drawn from MD are supported by experiments.

This is a revised version of a previously submitted manuscript, and both the MD simulations and experimental characterization of the novel SERT inhibitor a very thorough. I do have one question. The authors identify 3 poses for Lu AF60097 in the S2 site, termed I, II and core, and they selected core as the likely pose based on MM/GBSA calculations. While the difference in binding energy between I and core is quite large (nearly 18 kcal/mol), the difference of 3.9 kcal/mol between core and II is not very large considering the limited accuracy of MM/GBSA calculations. The authors validate the proposed binding pose through mutagenesis. However, I was unable to find a figure showing the alternate poses (especially pose II). Such a figure would be useful in order to assess whether the mutagenesis experiments convincingly rule out pose II as a possibility.

Reviewer #3:

Remarks to the Author:

The manuscript by Plenge et al and Loland explores an allosteric binding site and mechanism for SERT ligands. The SSRI S-citalopram acts as a strong positive allosteric modulator (PAM) of itself, and a much weaker one for the tricyclic antidepressant imipramine. The authors argue that this reflects conformational changes in SERT, exploring these with MD, and ultimately testing these predictions by mutagenesis. They go on to explore citalopram analog as potentially stronger PAMs, discovering a previously known analog, Lu AF60097, which has a 31 nM EC50 in the PAM, extracellular vestibule site for SERT. The binding of Lu AF60097, its interactions with imipramine (for which it is the better PAM, interestingly enough) and its effect on 5HT tone in vivo, are subsequently explored.

I found this story compelling, and certainly support publication. It's a great mixture of modeling, MD, allosteric communication, and even new ligand discovery, with in vivo impact.

The one weakness I saw was that there was no traditional allostery analysis. In GPCR pharmacology, for instance, it is typical to analyze allosteric effects via the alpha- and beta-coefficients, which are derived from concentration-response studies of the orthosteric molecule (imipramine, for instance) at different concentrations of the PAM (here, Lu AF60097). A PAM with a strong alpha-value will shift the CRCs to the left, one with a strong Beta value will increase the efficacy plateau, and molecules with both effects are certainly possible. That analysis is absent here, perhaps because of complications through binding at the second (primary) site on the part of Lu AF60097, but it is an important gap. Only with such formal allosteric analysis can one see saturation of the PAM effect (which should occur if it is genuinely a PAM), and knowing the alpha and beta effects can illuminate the molecular pharmacology and even biophysics of the molecule and its interactions. Perhaps a second point would be that, with two binding sites, one should see characteristic effects on the DRCs of Lu AF60097 itself-are those seen?

Notwithstanding these critiques, I overall support the publication of this manuscript.

Reviewers' comments:

Replies to reviewer's comments are denoted as point by point below.

Reviewer #1 (Remarks to the Author):

This manuscript focuses on the ability of some ligands for the serotonin transporter (SERT) to bind in the external vestibule of the transporter in addition to their primary site of binding at the locus where substrate (serotonin) binds. A second binding site was originally invoked to explain why high concentrations of substrate and some inhibitors could slow down dissociation of bound radioligands. The appearance of a second molecule of bound citalopram in the extracellular vestibule of SERT X-ray structures provided further encouragement for those attracted to the idea that transporters in this family had evolved an additional site for substrate and inhibitors in the service of some ill-defined physiological function. The alternative explanation for ligands to bind in this location is that it contains a mixture of polar and non-polar residues that interact to seal the external vestibule (EV) closed and that ligands present at high concentrations just stick there.

Irrespective of why this site exists, it is a shiny object that attracts attention, and is the subject of this manuscript. The authors have examined the interaction between a molecule of citalopram bound in this EV site and citalopram or imipramine bound in the high-affinity site. The two ligands are separated by about 6Å in the structure, and several side chains interact with both ligands. The first half of the work reported here concerns the interaction between the two molecules. The authors find that citalopram affinity in the EV site is higher when imipramine is bound at the substrate site than when citalopram is bound there. This interaction can be modulated by mutation of some residues whose side chains contribute to the regions common to the two binding sites. The observations are quite consistent with the structures and the computational modeling and mutagenesis support the authors' conclusions.

The remainder of the manuscript is devoted to the identification of an inhibitor from the Lundbeck collection of citalopram derivatives, which has higher affinity for the EV site than other inhibitors. However, this affinity, like that of citalopram, is higher when imipramine is bound at the substrate site. The authors model the derivative, Lu AF60097, into the EV site, and validate the binding pose with mutagenesis. They show that it binds both to the substrate site and the EV site and that the affinity to the EV site with imipramine in the substrate site is even higher than its affinity for the (empty) substrate site.

We thank the reviewer for this very positive outline of our findings.

These results predict that Lu AF60097 should inhibit serotonin transport synergistically with imipramine and this is observed in vitro. However, the only in vivo experiment shown is with Lu AF60097 alone. It inhibits, but why wasn't the synergy with imipramine tested?

The purpose of all this effort, purportedly, is to take advantage of the EV site to develop therapeutic drugs that would enhance the potency of existing drugs. If this ability to enhance

imipramine was not tested, it should have been. If it was, the results should be included.

We agree that this is an important point. We find that the therapeutic perspectives of an allosteric inhibitor is two-fold: (1) As nicely outlined by the reviewer to enhance the efficacy of an existing orthosteric ligand, and (2) as monotherapy to obtain higher selectivity than orthosteric ligands and possibly possessing different therapeutic indications. We completely agree with the reviewer, that the suggested experiment is an obvious additional choice to include in this manuscript. Unfortunately, the laboratory, which provided the initial microdialysis experiments, was no longer available. It took considerable time for the new laboratory to implement the setup and expertise that matched the previous. Hence this regrettably long resubmission period. We are sorry to keep you waiting for this essential experiment to be completed. Fortunately, we are now able to provide the results as new Figure 6b, which we find complement our *in vitro* results in Fig. 5c. As the *in vitro* experiments, our findings with *in vivo* microdialysis show no significant effect on extracellular 5-HT levels when IMI or Lu AF60097 are administered alone. Only when co-administered, we can measure a significant increase in 5-HT levels. The cumulative effect of 5-HT levels are shown as inset in Fig 6b.

We have inserted a description of the data in a new paragraph in Results section on page 15. We have also changed figure legend and updated the Methods section accordingly (see on page 29 and 24, respectively). We have also inserted an extra paragraph in the Discussion (page 18).

The point of this manuscript seems to be the therapeutic potential of EV ligands. However, the proof of principle *in vivo* seems missing. The other point made in the beginning of the manuscript is that ligand binding at the two adjacent sites can influence each other. I don't see this as very surprising, and it may be a bit over-sold as in the statement that binding in the EV site "is associated with robust conformational rearrangements". Conformational rearrangements in transporters implies changes that convert open-out to open-in forms. In this instance, they are side chain movements. The documentation of these movements and their consequences seems quite solid, but the results are not surprising and do not, in my mind, represent a fundamental increase in our understanding of ligand binding in SERT.

We do agree with the reviewer that it would be logic to suggest an allosteric interaction between the two sites. However, this is not what have been reported in the literature so far. Here the dissociation inhibition by a S2-bound ligand seems more to be due to a simple plug-the-hole mechanism. All crystal structures of S1-bound ligands to dDAT have shown the same constellation of the EV (Penmatsa et al. 2013, Nature; Penmatsa et al. 2015, Nat.Struc.Mol.Biol.; Wang et al. 2015, Nature). A similar observation is reported for solved SERT structures, where the S2-bound S-CIT did not change the EV structure relative to S1-bound S-CIT alone (Coleman et al. 2016, Nature). Only the structure of SERT in complex with ibogaine, show a different EV structure. Our previous MD simulations of SERT with S1-bound S-CIT and either C-CIT or clomipramine in S2 did also not show any allosteric mechanism between the two sites (Plenge et al. 2012, JBC). Accordingly, although conformational changes must happen in the S2 site during substrate transport, we do not find it a truism that two S1 inhibitors will have different impact on the EV.

Minor points:

The second sentence in the last paragraph on page 7 is unclear to me. It begins by describing the gauche rotamer of T497 when citalopram is bound at the substrate site and contrasts that to when imipramine is bound at the substrate site, which forces T497 into... the gauche rotamer?

We apologize for the confusion. What we meant is that the χ_1 rotamer of T497 is in *gauche+* in the S1:IMI/S2:apo condition, whereas in the S1:IMI/S2:S-CIT condition it is forced to go to *gauche-*. We have now clarified it on page 7.

Figure 2F is unclear. Here the conformation of the T497 side chain seems to change, unlike the text description.

We again apologize for the confusion. We have now clarified it in the main text (page 7) – the χ_1 rotamer of T497 goes from *gauche+* in the S1:IMI/S2:apo condition to *gauche-* in the S1:IMI/S2:S-CIT condition.

Why does almost every observation have to be "striking"?

We were/are indeed excited about findings. There are two instances that we used “strikingly”, the first one on page 8 is the improved inhibitory potency of S2:S-CIT in the presence of the T497A mutation. Such a gain of affinity is relatively rare in the mutagenesis studies in general and the only one in this study, and is a strong validation our structural hypothesis (see text). The second one on page 9 is about the drastically improved potency of the new analogs, which demonstrate that extracellular vestibule can be targeted for drug/ligand development reaching nanomolar range. We have now revised the first one to “remarkably”.

In the first sentence on page 9, it's not clear what a "more optimal" configuration would be. Please describe what is meant here.

What we meant by “more optimal” is the delicate spatial coordination that the Thr497-Phe335 motif that is required for high-affinity S2-ligands. The sentence has now been revised to: “Based on these findings, we hypothesized that the binding of S2-ligands having higher allosteric potency in the presence of S1:IMI must not result in steric crowdedness near the Thr497-Phe335 motif, while forming a favored interaction with Glu494.”

In the next paragraph, the sentence beginning "Interestingly," is overly complex. Try splitting into two smaller sentences.

We have now split the sentence into two.

The Y axis in Figure 5c is labeled Inhibition of uptake but looks more like uptake.

We agree. The typo is now corrected.

Reviewer #2 (Remarks to the Author):

This manuscript presents a very thorough characterization of a novel high affinity allosteric inhibitor of SERT. As SERT is the target of many psychoactive drugs and no compounds with high affinity for the S2 site have been previously reported, the potential significance of this work is high.

We thank the reviewer for this compliment.

the authors investigate the basis of allosteric coupling between the S1 and S2 sites primarily using molecular dynamics simulations, although the conclusions drawn from MD are supported by experiments.

This is a revised version of a previously submitted manuscript, and both the MD simulations and experimental characterization of the novel SERT inhibitor a very thorough. I do have one question. The authors identify 3 poses for Lu AF60097 in the S2 site, termed I, II and core, and they selected core as the likely pose based on MM/GBSA calculations. While the difference in binding energy between I and core is quite large (nearly 18 kcal/mol), the difference of 3.9 kcal/mol between core and II is not very large considering the limited accuracy of MM/GBSA calculations. The authors validate the proposed binding pose through mutagenesis. However, I was unable to find a figure showing the alternate poses (especially pose II). Such a figure would be useful in order to assess whether the mutagenesis experiments convincingly rule out pose II as a possibility.

We thank the reviewer for bringing up this issue. We have now inserted a new Figure S3 that shows all three Lu AF60097 binding poses in the extracellular vestibule. We further clarified about why we chose pose “core” at the end of page 9-10, “Consistent with the predicted binding free energy, the S-CIT core of Lu AF60097 in pose “I” protrudes out of the EV and is not fully engaged with hSERT, resulting in drastically weaker binding (Fig. S3a). Whereas pose “core” forms the ionic interaction with Glu494 and is in proximity to Lys490, pose “II” does not form interactions with either of these residues but a polar interaction with Asp328 (Fig. S3b,c). Therefore, we chose the pose “core” for further analysis because of its lower binding free energy and its ionic interaction with Glu494 (Fig. 3d,e). Our mutagenesis results of Asp328, Lys490, and Glu494 indeed support pose “core” but not pose “II” (see below)”, and a sentence on page 12, “This result also argues that Lu AF60097 is less likely to be in pose “II”, which forms a polar interaction with the sidechain of Asp328 (Fig. S3b).”

Reviewer #3 (Remarks to the Author):

The manuscript by Plenge et al and Loland explores an allosteric binding site and mechanism for SERT ligands. The SSRI S-citalopram acts as a strong positive allosteric modulator (PAM) of itself, and a much weaker one for the tricyclic antidepressant imipramine. The authors argue that this reflects conformational changes in SERT, exploring these with MD, and ultimately testing these predictions by mutagenesis. They go on to explore citalopram analogs as potentially stronger PAMs, discovering a previously known analog, Lu AF60097, which has a 31 nM EC50 in the PAM, extracellular vestibule site for SERT. The binding of Lu AF60097, its interactions with imipramine (for which it is the better PAM, interestingly enough) and its effect on 5HT tone *in vivo*, are subsequently explored.

I found this story compelling, and certainly support publication. It's a great mixture of modeling, MD, allosteric communication, and even new ligand discovery, with *in vivo* impact.

We thank the reviewer for this positive opinion on our efforts and findings.

The one weakness I saw was that there was no traditional allosteric analysis. In GPCR pharmacology, for instance, it is typical to analyze allosteric effects via the alpha- and beta-coefficients, which are derived from concentration-response studies of the orthosteric molecule (imipramine, for instance) at different concentrations of the PAM (here, Lu AF60097). A PAM with a strong alpha-value will shift the CRCs to the left, one with a strong Beta value will increase the efficacy plateau, and molecules with both effects are certainly possible. That analysis is absent here, perhaps because of complications through binding at the second (primary) site on the part of Lu AF60097, but it is an important gap. Only with such formal allosteric analysis can one see saturation of the PAM effect (which should occur if it is genuinely a PAM), and knowing the alpha and beta effects can illuminate the molecular pharmacology and even biophysics of the molecule and its interactions.

We agree that an adoption of allosteric analysis from the GPCR field would have been beneficial here. However, as the reviewer nicely points out, it is not possible to retrieve any alpha- or beta-coefficients of Lu AF60097 by its effect on [³H]IMI binding. The reason for this could be the association of Lu AF60097 to the primary (S1) binding site. Another possibility is that since [³H]IMI only dissociates from the S1 site in the absence of bound Lu AF60097, or any other S2-bound ligand, we will only be able to measure the affinity for [³H]IMI when it is not allosterically affected by the S2-bound ligand. At least this allosteric interaction is different from what is observed in GPCRs.

We respectfully disagree that our method using dissociation inhibition does not show saturation of the PAM effect. Our experiments correlate the dissociation inhibition of the S1-bound ligand with the added concentration of the S2 bound ligand. The bottom of the allosteric potency curve is reflecting a situation when no dissociation occurs of the S1-ligand and, hence, the S2 site is saturated. An example of the data that underlie an allosteric potency curve in the manuscript is shown here for clarification:

Dissociation inhibition of [³H]S-citalopram by increasing concentrations [0.8 – 200 μM] of unlabeled S-Citalopram (S-Cit)

The data in the left graph converted to reflect the allosteric potency

The used method for measurement of dissociation inhibition, as an indication of allosteric interaction, is the ‘golden standard’ for NSSs, see e.g. Navranta et al 2018, PLoS One; Larsen et al. 2016, Br J Pharmacol; Rothman et al. 2015, JPET; Chen et al. 2005 Eur Neuropharmacol; Wennogle & Meyerson 1982, Eur J Pharmecol.

Perhaps a second point would be that, with two binding sites, one should see characteristic effects on the DRCs of Lu AF60097 itself--are those seen?

Possibly, but it would require radiolabeled Lu AF60097 to perform the experiment. This is not available.

Notwithstanding these critiques, I overall support the publication of this manuscript.

Thank you so much. We appreciate it.

Reviewers' Comments:

Reviewer #1:

None

Reviewer #2:

Remarks to the Author:

This is a revised version of a manuscript previously reviewed. As far as I am concerned, the authors have addressed my previous comments and concerns.

Reviewer #3:

None.